# Dynamic clustering of dynamin-amphiphysin helices regulates membrane constriction and fission coupled with GTP hydrolysis

**Tetsuya Takeda[1]\*, Toshiya Kozai[2], Huiran Yang[1], Daiki Ishikuro[2], Kaho Seyama[1], Yusuke Kumagai[2], Tadashi Abe[1], Hiroshi Yamada[1], Takayuki Uchihashi[3,4], Toshio Ando[3,5]\*, Kohji Takei[1,3]\***

[1]Graduate School of Medicine, Dentistry and Pharmaceutical Sciences, Okayama University, Okayama, Japan; [2]Department of Physics, College of Science and Engineering, Kanazawa University, Kanazawa, Japan; [3]CREST, JST, Saitama, Japan; [4]Department of Physics, School of Science, Nagoya University, Nagoya, Japan; [5]Bio-AFM Frontier Research Center, College of Science and Engineering, Kanazawa University, Kanazawa, Japan

**Abstract** Dynamin is a mechanochemical GTPase essential for membrane fission during clathrin-mediated endocytosis. Dynamin forms helical complexes at the neck of clathrin-coated pits and their structural changes coupled with GTP hydrolysis drive membrane fission. Dynamin and its binding protein amphiphysin cooperatively regulate membrane remodeling during the fission, but its precise mechanism remains elusive. In this study, we analyzed structural changes of dynamin-amphiphysin complexes during the membrane fission using electron microscopy (EM) and high-speed atomic force microscopy (HS-AFM). Interestingly, HS-AFM analyses show that the dynamin-amphiphysin helices are rearranged to form clusters upon GTP hydrolysis and membrane constriction occurs at protein-uncoated regions flanking the clusters. We also show a novel function of amphiphysin in size control of the clusters to enhance biogenesis of endocytic vesicles. Our approaches using combination of EM and HS-AFM clearly demonstrate new mechanistic insights into the dynamics of dynamin-amphiphysin complexes during membrane fission.

DOI: https://doi.org/10.7554/eLife.30246.001

**\*For correspondence:**
ttakeda@okayama-u.ac.jp (TT);
tando@staff.kanazawa-u.ac.jp (TA);
kohji@md.okayama-u.ac.jp (KT)

**Competing interests:** The authors declare that no competing interests exist.

## Introduction

Clathrin-mediated endocytosis (CME) is the best characterized endocytic pathway by which cells incorporate extracellular molecules into cells as cargoes of clathrin-coated vesicles (*Kirchhausen et al., 2014*; *McMahon and Boucrot, 2011*). CME is required for various essential processes including neuronal transmission, signal transduction and other cell membrane activities such as cell adhesion and migration. For precise progression of membrane invagination and fission during CME, various proteins need to be assembled in a temporally and spatially coordinated manner at the site of endocytosis.

One of those endocytic proteins, dynamin, is a GTPase essential for membrane fission in CME (*Antonny et al., 2016*; *Ferguson and De Camilli, 2012*; *Schmid and Frolov, 2011*). There are three dynamin isoforms in mammals: dynamin 1 and dynamin 3, two tissue-specific isoforms which are highly expressed in neurons, and dynamin 2, an ubiquitously expressed isoform (*Cao et al., 1998*; *Cook et al., 1996*, *1994*). Structural studies from several groups demonstrated that dynamin consists of five structurally distinct domains: a GTPase domain, a bundle signaling element (BSE), a stalk,

**eLife digest** The nerve cells that make up a nervous system connect at junctions known as synapses. When a nerve impulse reaches the end of the cell, membrane-bound packages called vesicles fuse with the surface membrane and release their contents to the outside. The contents, namely chemicals called neurotransmitters, then travels across the synapse, relaying the signal to the next cell.

Nerve cells can fire many times per second. The membrane from fused vesicles must be retrieved from the surface membrane and recycled to make new vesicles, ready to transmit more signals across the synapse. Many proteins at these sites are involved in folding the fused membrane back into the cell, constricting the opening, and eventually pinching off the new vesicles – a process known as endocytosis. Two proteins named dynamin and amphiphysin cooperate in this process, but their precise mechanism remained elusive. Dynamin is a protein that acts like a motor; it breaks down a molecule called GTP to release energy. Previous studies have seen that dynamin-amphiphysin complexes join end to end to form long helical structures.

Takeda et al. have now looked at how the structure of the helices changes during endocytosis. This revealed that the dynamin-amphiphysin helices rearrange to form clusters when the GTP is broken down. Further analysis showed that the folded membrane becomes constricted at regions that are not coated with the clusters of dynamin-amphiphysin helices. Takeda et al. also discovered that amphiphysin controls the size of the clusters to help make the new vesicles more uniform.

The gene for dynamin is altered in a number of disorders affecting the nervous system and muscles, including epileptic encephalopathy, Charcot-Marie-Tooth disease and congenital myopathy. Moreover, a neurological disorder characterized by muscle stiffness (known as Stiff-person syndrome) occurs when an individual's immune system mistakenly attacks the amphiphysin protein. As such, these new findings will not only help scientists to better understand the process of endocytosis, but they will also give new insight into a number of human diseases.

DOI: https://doi.org/10.7554/eLife.30246.002

a pleckstrin homology (PH) domain and a proline-rich domain (PRD) from N-terminus to C-terminus (*Faelber et al., 2011*; *Ford et al., 2011*; *Reubold et al., 2015*). The GTPase domain is responsible for hydrolysis of GTP (guanosine triphosphate) and the PH domain is required for membrane association by binding to negatively charged phospholipids such as PI (4,5) $P_2$ (phosphatidylinositol 4,5-bisphosphate). The stalk structure serves as a binding interface for dimerization and oligomerization of dynamin. The BSE, which is located between the stalk and GTPase domain, functions as a flexible hinge required for structural changes of dynamin upon GTP hydrolysis. Dynamin forms helical oligomers which was first observed in presynaptic terminals of *shibire* mutant flies at restrictive temperature (*Koenig and Ikeda, 1989*). Dynamin also assembles into helices at the neck of endocytic pits in the isolated presynaptic nerve terminals treated with slowly hydrolyzable GTP analogue GTPγS (guanosine 5'-O-[gamma-thio]triphosphate) (*Takei et al., 1995*). Similar dynamin helices were reconstituted in vitro either with liposomes (*Sweitzer and Hinshaw, 1998*; *Takei et al., 1998*) or without liposomes in a low-salt condition (*Hinshaw and Schmid, 1995*).

There is a consensus view about the dynamin-mediated membrane constriction and fission which is well supported by previous studies from different groups: membrane constriction is required, but not sufficient, for fission (*Antonny et al., 2016*; *Faelber et al., 2012*; *Schmid and Frolov, 2011*). However, it is still controversial how constriction is achieved, and what GTP energy is used for. For example, membrane constriction could be achieved by assembly into the highly constricted state when dynamin is bound to GTP (*Chen et al., 2004*; *Mattila et al., 2015*; *Mears et al., 2007*; *Zhang and Hinshaw, 2001*). Alternatively, membrane constriction could be achieved by hydrolysis of GTP that induces a conformational change leading to constriction (*Cocucci et al., 2014*; *Marks et al., 2001*; *Roux et al., 2006*). However, precise mechanisms involved in dynamin-mediated membrane constriction and fission remain unclear.

Amphiphysin is a BAR domain protein required for membrane invagination in CME (*Wigge et al., 1997*). Amphiphysin has a lipid interacting BAR (Bin–Amphiphysin–Rvs) domain in its N-terminal, a medial clathrin/AP-2 binding (CLAP) domain and C-terminal Src homology 3 (SH3) domain. The BAR

domain of amphiphysin forms crescent-shaped dimer and its concave surface serves as a platform for bending membrane or sensing membrane curvature (*Peter et al., 2004*). The CLAP domain binds to clathrin and AP-2, major components of clathrin-coated pits, and helps to recruit amphiphysin to the sites of CME. In addition, the C-terminal SH3 domain of amphiphysin binds directly to the PRD of dynamin 1 (*David et al., 1996*; *Takei et al., 1999*) and enhances dynamin's GTPase activity in the presence of liposomes (*Takei et al., 1999*; *Yoshida et al., 2004*). Amphiphysin copolymerizes with dynamin 1 into helical complexes, which form membrane tubules in vitro (*Takei et al., 1999*; *Yoshida et al., 2004*) similar to those formed from synaptic plasma membranes (*Takei et al., 1995*). Furthermore, injection of specific antibodies against amphiphysin into the giant synapse in lampreys (*Evergren et al., 2004*) or amphiphysin KO in mice (*Di Paolo et al., 2002*) causes suppressed endocytosis in synaptic vesicle recycling. These results suggest that dynamin mediates membrane fission in CME in collaboration with amphiphysin in vivo. However, the precise contribution of amphiphysin in the dynamin-mediated membrane fission remains elusive.

In this study, we analyzed dynamics of dynamin-amphiphysin helical complexes using an approach combining electron microscopy (EM) and high-speed atomic force microscopy (HS-AFM). Firstly, we show that the dynamin-amphiphysin helices are rearranged to form clusters upon GTP hydrolysis, and membrane constriction occurs at protein-uncoated regions between the clusters. Secondly, we reveal that GTP hydrolysis is required and sufficient for the cluster formation by dynamin-amphiphysin complexes by EM analyses. Finally, we show a novel function of amphiphysin in controlling cluster size, which in turn regulates biogenesis of endocytic vesicles. These findings provide new insights into the mechanism of membrane constriction and fission by dynamin-amphiphysin complexes.

## Results

### GTP hydrolysis is required and sufficient for membrane constriction by dynamin-amphiphysin complexes

To elucidate the mechanisms of dynamin-mediated membrane fission, we reconstituted the minimum system in vitro and analyzed the time course of its structural changes using EM. Human dynamin 1 and amphiphysin were purified (*Figure 1—figure supplement 1A*) and their activity to form ring-shaped complexes in a buffer of physiological ionic strength and pH condition (*Figure 1—figure supplement 1B*) at different stoichiometry of dynamin and amphiphysin (*Figure 1—figure supplement 2A*) were confirmed. As previously described (*Sweitzer and Hinshaw, 1998*; *Takei et al., 1999*), the dynamin-amphiphysin complexes induced tubulation of large unilamellar vesicles (LUVs) in the absence of GTP (*Figure 1A*, No GTP) and it is not stoichiometry dependent (*Figure 1—figure supplement 2B*). Immediately after the addition of 1 mM GTP, the appearance of lipid tubules was not affected (*Figure 1A*, GTP 1 s), but they started to form multiple constriction sites over time (*Figure 1A*, GTP 5 s, 10 s and 30 s) and membrane fission occurred finally and numerous vesicles were generated within 1 min (*Figure 1A*, GTP 1 min). The membrane fission activity by dynamin-amphiphysin complexes was stoichiometry sensitive: the membrane fission occurred efficiently at 1:0.5 or 1:1 molar ratio of dynamin and amphiphysin (*Figure 1-figure supplement 2C*, 1:0.5 and 1:1), while the fission activity was less efficient when dynamin and amphiphysin were mixed at 1:2 ratio (*Figure 1-figure supplement 2C*, 1:2), probably due to lower stimulatory effect of amphiphysin on dynamin GTPase activity at the higher molecular ratio (*Yoshida et al., 2004*).

Next, we tried to clarify how the membrane constriction and fission by dynamin-amphiphysin helices are correlated with guanine nucleotide conditions during GTP hydrolysis. The appearance of lipid tubules (*Figure 1B*, No GTP) was not affected in the presence of either slowly hydrolyzable GTP analogue GTPγS (*Figure 1B*, GTPγS) or non-hydrolyzable GTP analogue GMP-PNP (guanosine 5′-[β,γ-imido]triphosphate) (*Figure 1B*, GMP-PNP). In contrast, in the presence of GDP (guanosine diphosphate) and vanadate, the complex which mimics the GDP·Pi transition state, lipid tubules were constricted at multiple sites (*Figure 1B*, GDP + vanadate). Addition of only GDP did not cause membrane constriction or fission, but membrane tubules were deformed (*Figure 1B*, GDP). Finally, numerous vesicles were generated 10 min after the addition of 1 mM GTP, in which multiple rounds of GTP hydrolysis were likely to have taken place (*Figure 1B*, GTP). Taken these results together, GTP hydrolysis is essential for both membrane constriction and fission by the dynamin-amphiphysin

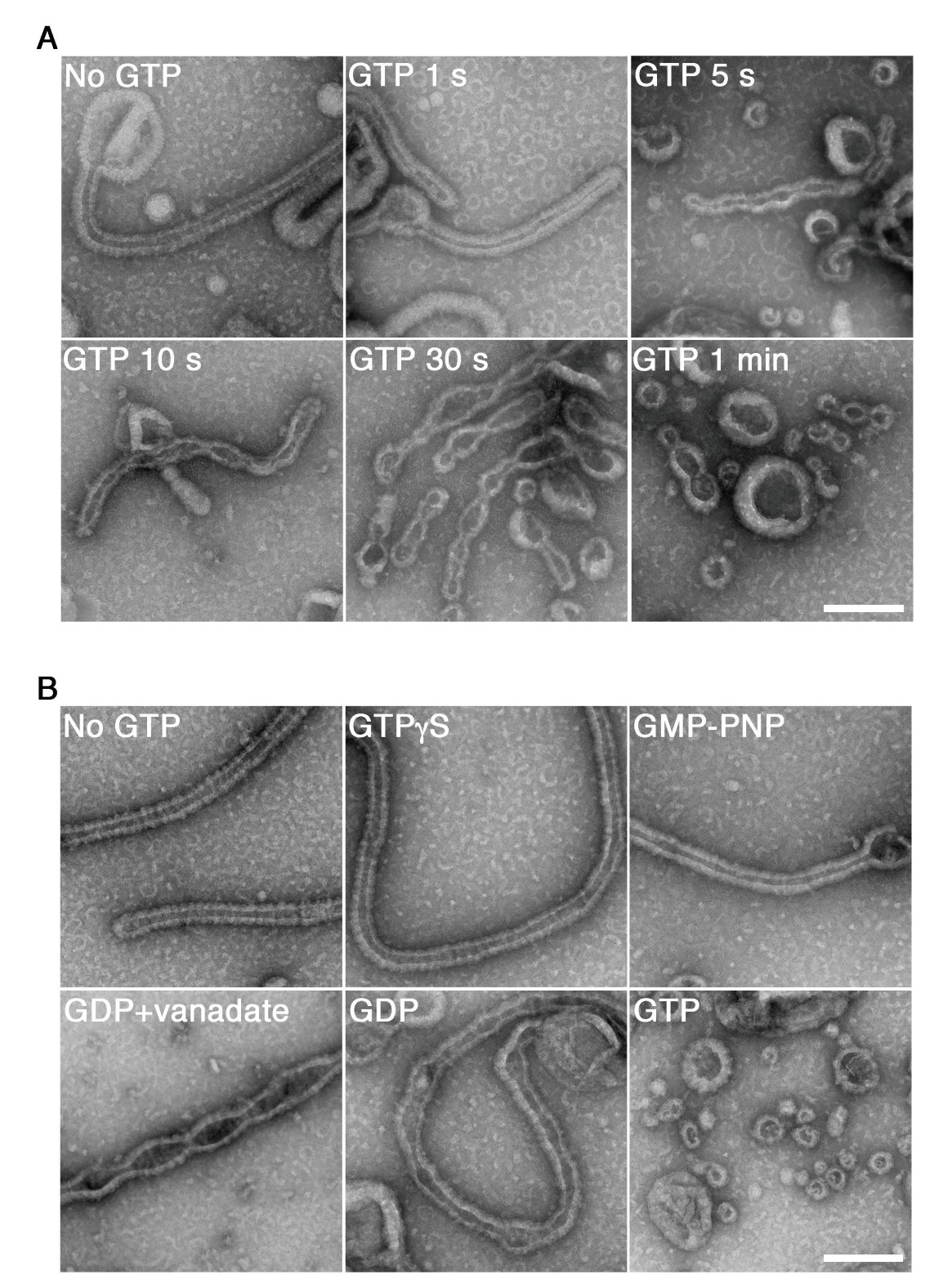

**Figure 1.** GTP hydrolysis is required and sufficient for membrane constriction by dynamin-amphiphysin helical complexes. (A) Electron micrographs of lipid tubules induced by dynamin-amphiphysin helical complex before GTP addition (No GTP) and at different time points after addition of 1 mM GTP (GTP 1 s, GTP 5 s, GTP 10 s, GTP 30 s and GTP 1 min). More than thirty samples from three individual experiments were examined and representative images are shown. Scale bar is 200 nm. (B) Electron micrographs of lipid tubules induced by dynamin-amphiphysin helical complex without guanine

*Figure 1 continued on next page*

*Figure 1 continued*

nucleotide (No GTP) or with a transition states analogue of GTPase reaction, by adding 1 mM each of slowly hydrolyzable GTP analogue (GTPγS), nonhydrolyzable GTP analogue (GMP-PNP), GDP combined with vanadate (GDP + vanadate), GDP (GDP), or GTP (GTP) for 10 min. More than thirty samples from three individual experiments were examined and representative images are shown. Scale bar is 200 nm.

DOI: https://doi.org/10.7554/eLife.30246.003

The following figure supplements are available for figure 1:

**Figure supplement 1.** Purified dynamin and amphiphysin forms ring-shaped complexes.

DOI: https://doi.org/10.7554/eLife.30246.004

**Figure supplement 2.** Stoichiometry dependency of dynamin and amphiphysin in ring complex formation, liposome tubulation and membrane fission.

DOI: https://doi.org/10.7554/eLife.30246.005

complexes, but subsequent dissociation of GTP hydrolytic products (GDP and/or phosphate) is required for completing membrane fission.

## GTP hydrolysis induces clustering of dynamin-amphiphysin helices

Although we determined the requirement of GTP hydrolysis in membrane constriction and fission by the dynamin-amphiphysin complexes, structural changes of the complexes were not clearly resolved in the in vitro assay system using LUVs. To improve the resolution, we used rigid lipid nanotubes containing glycolipid galactosylceramide (GalCer) (*Wilson-Kubalek et al., 1998*), instead of using LUVs in the in vitro assay system. Lipid nanotubes are rod-shaped liposomes and similar in size to the unconstricted necks of clathrin-coated pits observed in vivo (*Figure 2A*, Nanotube). Dynamin-amphiphysin complexes assembled into helices on the lipid nanotubes (*Figure 2A*, No GTP), which is similar to those formed by dynamin alone (*Stowell et al., 1999*). Interestingly, the dynamin-amphiphysin helices transiently formed clusters after the addition of GTP (*Figure 2A*, GTP 1 s and 20 s, brackets). The dynamin-amphiphysin clusters were disorganized over time and partially dissociated from the nanotubes (*Figure 2A*, GTP 30 s and 1 min).

To correlate the dynamics of dynamin-amphiphysin complexes with GTP hydrolysis, we examined structural changes of the complexes on lipid nanotubes at different transition states of GTP hydrolysis. The appearance of dynamin-amphiphysin helices was unchanged even in the presence of GTPγS or GMP-PNP (*Figure 2B*, No GTP, GTPγS and GMP-PNP). Interestingly, addition of GDP and vanadate induced rearrangement of the dynamin-amphiphysin helical complexes to form clusters similar to those observed after the addition of GTP (*Figure 2B*, GDP + vanadate). The average pitch of helices in the clusters were shorter (15.0 ± 0.3 nm, mean pitch ± s.e.m.) compared to the average pitch of the helical complexes in No GTP control (20.0 ± 0.5 nm, mean pitch ± s.e.m.) (*Figure 2—source data 1*). Furthermore, unlike membrane fission activity (*Figure 1—figure supplement 2C*), clustering behavior of the dynamin-amphiphysin complexes on the lipid nanotubes were not stoichiometry dependent (*Figure 2—figure supplement 1*, 1:0.5, 1:1 and 1:2, white brackets). In contrast, GDP alone did not affect the distribution of dynamin-amphiphysin helices (*Figure 2B*, GDP). Finally, the dynamin-amphiphysin helical complexes were disorganized and eventually dissociated from the lipid nanotubes 10 min after the addition of 1 mM GTP (*Figure 2B*, GTP). Taken these results together, the dynamin-amphiphysin helical complexes transiently form clusters in the GTP hydrolysis transition state of GDP·Pi during which membrane tubules are constricted.

## Dynamic clustering of dynamin-amphiphysin helical complexes upon GTP hydrolysis is revealed by HS-AFM

To elucidate the dynamics of dynamin-amphiphysin helical complexes during the membrane constriction and fission, we analyzed the clustering process of the complexes using HS-AFM (*Ando et al., 2013*). LUVs were stably immobilized on the carbon-coated and glow-discharged mica substrate (*Figure 3—figure supplement 1A*; *Video 1*), and they were successfully tubulated in the presence of dynamin and amphiphysin (*Figure 3—figure supplement 1B*; *Video 2*). The dynamin-amphiphysin helices on the lipid tubules were aligned with an almost regular pitch (22.0 ± 0.7 nm, mean pitch ± s.e.m.) and they were immobile before GTP addition (*Figure 3A*, 0 s and 21 s; *Video 3*; *Figure 3—source data 1*). Interestingly, the dynamin-amphiphysin helices became mobile after GTP addition and eventually formed clusters consisting of a few helices with shorter pitch (15.7 ± 0.3 nm,

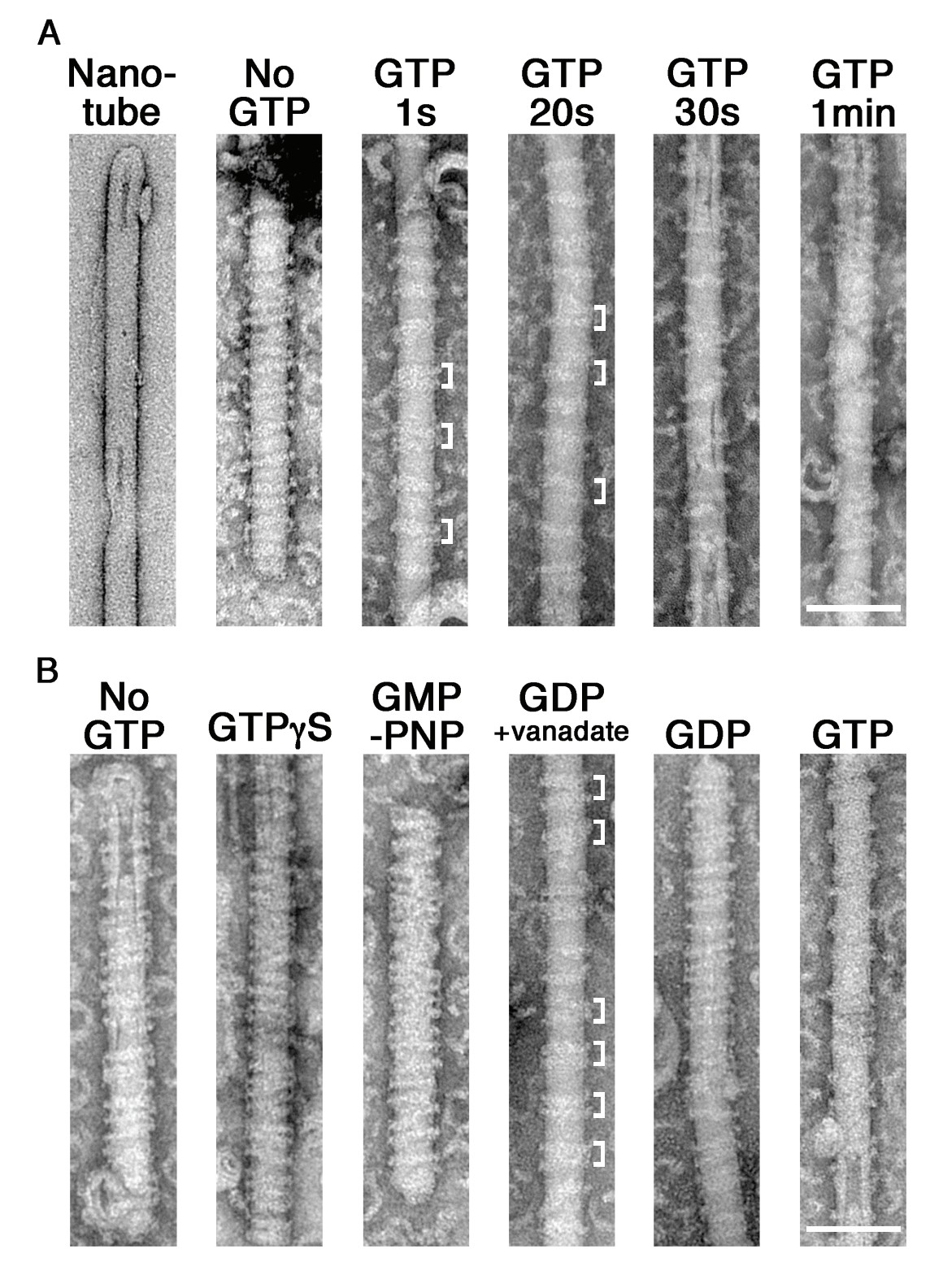

**Figure 2.** GTP hydrolysis induces clustering of dynamin-amphiphysin complexes on lipid nanotubes. (**A**) Electron micrographs of a lipid nanotube (Nanotube) and those with dynamin-amphiphysin complexes before GTP addition (No GTP) and at different time points after GTP addition (GTP 1 s, GTP 20 s, GTP 30 s, and GTP 1 min). Clusters of dynamin-amphiphysin helical complexes are indicated (white brackets). More than thirty samples from three individual experiments were examined and representative images are shown. Scale bar is 100 nm. (**B**) Electron micrographs of lipid nanotubes

*Figure 2 continued on next page*

*Figure 2 continued*

after addition of dynamin-amphiphysin complexes without guanine nucleotide (No GTP) or with a transition states analogue of GTPase reaction, by adding 1 mM each of slowly hydrolysable GTP analogue (GTPγS), nonhydrolyzable GTP analogue (GMP-PNP), GDP combined with vanadate (GDP + vanadate), GDP (GDP) and GTP (GTP) for 10 min. More than 30 samples from three individual experiments were examined and representative images are shown. Clusters of dynamin-amphiphysin helical complexes are indicated (white brackets). The average pitch of helices in the clusters is $15.00 \pm 2.2$ nm (mean pitch ± s.e.m., n = 63 from 7 nanotubes) in GDP + vanadate, while the average pitch of the helical complexes is $20.0 \pm 0.5$ nm (mean pitch ± s.e.m., n=81 from 9 nanotubes) in No GTP control. Scale bar is 100 nm.

DOI: https://doi.org/10.7554/eLife.30246.006

The following source data and figure supplement are available for figure 2:

**Source data 1.** Measuring pitch size of dynamin-amphiphysin helices on the lipid nanotube without GTP and in the presence of GDP and vanadate in panel B.

DOI: https://doi.org/10.7554/eLife.30246.008

**Figure supplement 1.** Stoichiometry dependency of dynamin and amphiphysin in cluster formation

DOI: https://doi.org/10.7554/eLife.30246.007

mean pitch ± s.e.m.) (*Figure 3A*, from 42 s to 131 s; *Video 3*; *Figure 3—source data 1*). Particle tracking analyses of the individual dynamin-amphiphysin helices showed that the dynamin-amphiphysin complexes were static before GTP addition (*Figure 3B*, 5-21 s; *Video 4*), but addition of 1 mM GTP stimulated longitudinal movement of the helical complexes, leading to the cluster formation (*Figure 3B*, 38-54 s, 38–86 s and 38–118 s; *Video 5*). Although membrane fission was not observed in this sample probably due to a strong attachment of the lipid tubule to the substrate, the helices had a tendency to constrict during the cluster formation (*Figure 3C*; *Figure 3—figure supplement 2*; *Figure 3—source data 2*; *Figure 3—figure supplement 2—source data 1*). These results suggest that dynamin-amphiphysin helical complexes undergo two modes of structural changes, longitudinal clustering and radial constriction, during GTP hydrolysis.

## Membrane fission occurs at protein-uncoated regions flanking dynamin-amphiphysin clusters

We next tried to correlate the cluster formation of dynamin-amphiphysin helical complexes with membrane constriction and fission. In the representative sample in which membrane constriction and fission occurred, a few dynamin-amphiphysin helices merged to form a cluster over time after GTP addition (*Figure 4A*, 0 s, 125.3 s, 185.5 s and 227.5 s; *Video 6*; *Figure 4—source data 1*). Interestingly, membrane constriction occurred at flanking regions of the cluster where membrane was bare of dynamin-amphiphysin complexes (*Figure 4A*, fission point (FP).1 and FP.2). The heights at sites marked with FP.1 and FP.2 were not changed before constriction (*Figure 4B*, before constriction; *Figure 4—source data 2*), but they became lower in a stepwise manner from a pre-constriction height of around 30 nm down to 20–25 nm or below (*Figure 4B*, after constriction; *Figure 4—source data 2*). Similar longitudinal redistribution of the dynamin-amphiphysin helices before membrane constriction was also observed in another sample, in which constriction occurred at one end of clustered dynamin-amphiphysin complexes (*Figure 4c*, arrow; Video 8). These results strongly suggest that membrane constriction and fission occur at the protein-uncoated regions created as a result of the clustering of dynamin-amphiphysin helical complexes.

## Amphiphysin contributes to efficient vesicle formation by controlling cluster formation

We previously demonstrated that amphiphysin stimulates the GTPase activity of dynamin and thus enhances vesicle biogenesis (*Yoshida et al., 2004*). In this study, we also noticed that the average size of vesicles formed by dynamin-amphiphysin complexes ($70.0 \pm 2.9$ nm, mean diameter ± s.e.m.) was significantly smaller compared to those formed by dynamin alone ($204.6 \pm 12.3$ nm, mean diameter ± s.e.m.) after GTP addition (*Figure 5A*; *Figure 5—source data 1*). Consistently, dynamin-amphiphysin complex formed constriction sites with shorter intervals ($150.3 \pm 9.8$ nm, mean intervals ± s.e.m.) compared to those formed by dynamin alone ($193.5 \pm 15.8$ nm, mean intervals ± s.e.m.) in the presence of GDP and vanadate (*Figure 5B*; *Figure 5—source data 2*). To further elucidate roles of amphiphysin in the membrane constriction and fission, the cluster formation by dynamin alone was compared to that by dynamin-amphiphysin complexes, using lipid nanotubes. As

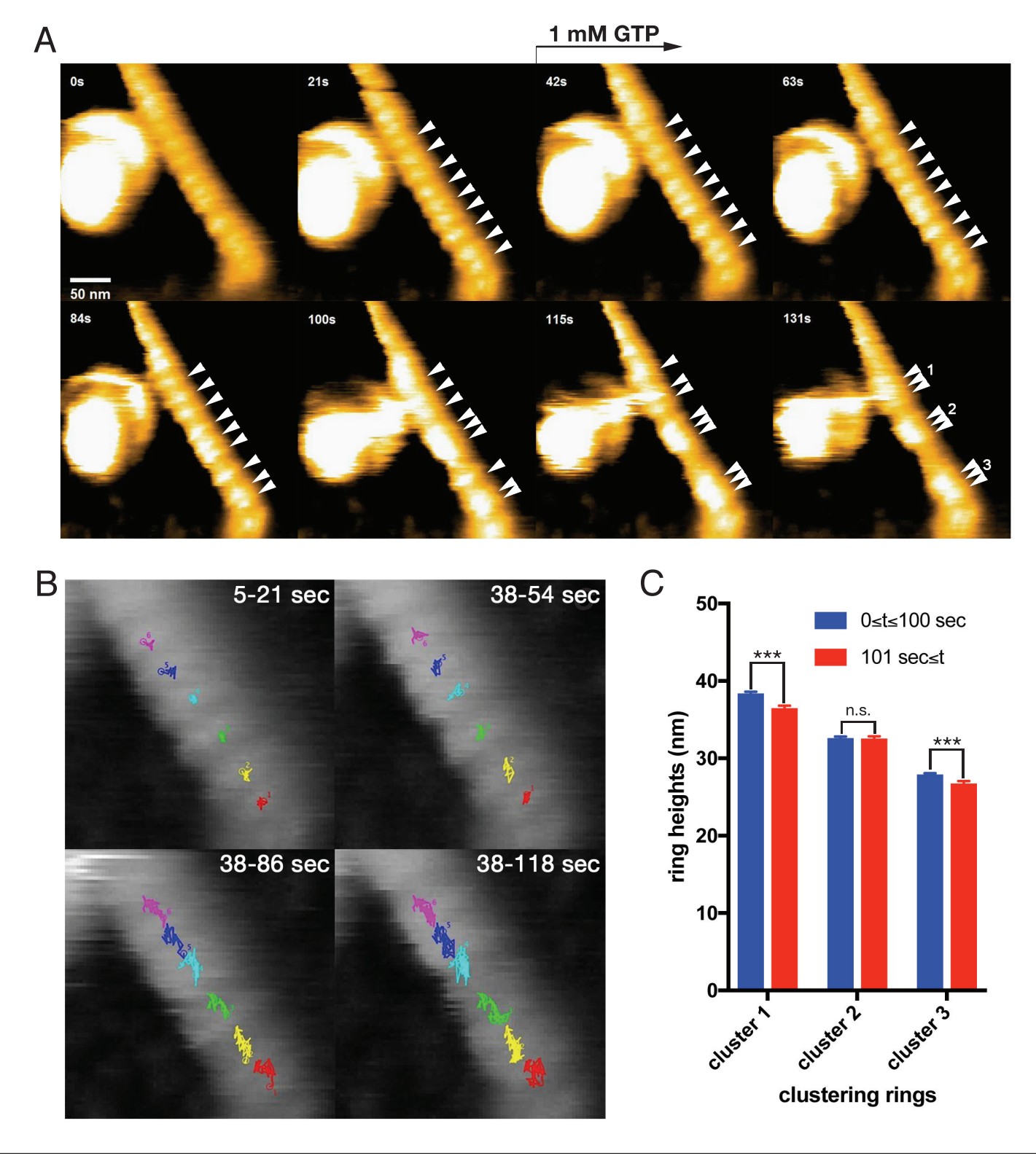

**Figure 3.** Dynamic clustering of dynamin-amphiphysin helices during GTP hydrolysis. (**A**) HS-AFM images captured at 1 frame/s of dynamin-amphiphysin helical complexes on membrane tubules before (0 s and 21 s) and after GTP addition at different time points (42 s, 63 s, 84 s, 100 s, 115 s and 131 s). Dynamin-amphiphysin helices (arrowheads) are assembled into three distinct clusters (1, 2 and 3 at 131 s). The pitch of dynamin-amphiphysin helices on the lipid tubule was 22.0 ± 0.7 nm (mean pitch ± s.e.m., n = 36 from 3 time points) before GTP addition and 15.7 ± 0.3 nm (mean pitch ± s.e.

*Figure 3 continued on next page*

*Figure 3 continued*

m., n = 36 from 9 time points) after GTP addition. (B) Particle tracking of dynamin-amphiphysin helices before (5–21 s) and after addition of 1 mM GTP (38–54 s, 38–86 s and 38–118 s) from *Videos 4* and *5*, respectively. Particle tracking of the complexes in the cluster 2 (light blue, dark blue and magenta) and cluster 3 (red, yellow and green) are shown. (C) Dynamin-amphiphysin helices tend to constrict during clustering. Average heights before ($0 \leq t \leq 100$ s) and after clustering ($101$ s $\leq t$) are 38.4 ± 0.2 nm and 36.5 ± 0.2 nm for cluster 1, 32.6 ± 0.2 nm and 32.6 ± 0.2 nm for cluster 2, 27.9 ± 0.1 nm and 26.8 ± 0.2 nm for cluster 3, respectively. The heights were measured from the substrate surface. The marks *** indicate p<0.001 and n.s. is not significant, respectively.

DOI: https://doi.org/10.7554/eLife.30246.009

The following source data and figure supplements are available for figure 3:

**Source data 1.** Measuring pitch size of dynamin-amphiphysin helices on the lipid tubule before and after cluster formation.
DOI: https://doi.org/10.7554/eLife.30246.012
**Source data 2.** Measuring heights of dynamin-amphiphysin helices on the lipid tubule before and after cluster formation.
DOI: https://doi.org/10.7554/eLife.30246.013
**Figure supplement 1.** HS-AFM imaging of LUV and its tubulation by dynamin-amphiphysin complex.
DOI: https://doi.org/10.7554/eLife.30246.010
**Figure supplement 2.** Dynaimin-amphiphysin helices constrict during cluster formation.
DOI: https://doi.org/10.7554/eLife.30246.011
**Figure supplement 2—source data 1.** Measuring heights of dynamin-amphiphysin helices on the lipid tubule during cluster formation.
DOI: https://doi.org/10.7554/eLife.30246.014

already described, dynamin-amphiphysin complexes formed clusters with a few helices in the presence of GDP and vanadate (34.2 ± 1.7 nm, mean cluster size ± s.e.m.) (*Figure 5C*, Dynamin + Amphiphysin; *Figure 5—source data 3*). In contrast, dynamin alone formed larger-sized clusters consist of more helical complexes (59.3 ± 4.7 nm, mean cluster size ± s.e.m.) (*Figure 5C*, Dynamin; *Figure 5—source data 3*). These results suggest that amphiphysin contributes to the effective generation of properly sized vesicles by controlling the cluster formation of dynamin-amphiphysin helical complexes.

## Discussion

In this study, we analyzed dynamics of dynamin-amphiphysin helical complexes during membrane constriction and fission using EM and HS-AFM. EM analyses showed that GTP hydrolysis is required for both membrane constriction and fission, but dissociation of hydrolytic products (GDP and/or phosphate) seems necessary for the completion of membrane fission (*Figure 1*). In the presence of GTP or GDP and vanadate, dynamin-amphiphysin helical complexes are reorganized, resulting in the formation of clusters consisting of a few dynamin-amphiphysin helices (*Figure 2*). HS-AFM analyses directly demonstrated that GTP hydrolysis induces dynamic longitudinal movement of the dynamin-amphiphysin helices as well as constriction during the cluster formation (*Figure 3*). Interestingly, HS-AFM analyses also demonstrated that membrane constriction and fission occur at the 'protein-uncoated' regions created as a result of cluster formation of dynamin-amphiphysin complexes (*Figure 4*). Finally, we found that amphiphysin contributes to effective biogenesis of endocytic vesicles by regulating size of the clusters formed by dynamin-amphiphysin helical complexes (*Figure 5*).

There is a consensus view about the requirement of GTP hydrolysis in membrane fission, but the requirement of GTP hydrolysis in membrane constriction is still controversial (*Antonny et al., 2016*). Membrane tubules are constricted in the presence of non-hydrolyzable GTP analogue (*Chen et al., 2004*; *Mears et al., 2007*; *Zhang and Hinshaw, 2001*) and more constricted with a GTP-loaded GTPase defective K44A mutant (*Sundborger et al., 2014*). In both cases, membrane tubules are evenly constricted and periodical membrane constriction sites which lead to membrane fission is not created. In the present study, we showed that membrane constriction sites are created in the presence of GDP and vanadate, which mimicked a transition state of GTP hydrolysis (GDP·Pi), suggesting that complete hydrolysis of GTP is required for the formation of constriction sites leading to membrane fission (*Figure 1B*). Membrane fission has never been observed in the presence of GDP and vanadate, suggesting that release of GTP hydrolytic products (GDP and/or phosphate) is a prerequisite for membrane fission. Further analyses will more precisely reveal which intermediate state in the

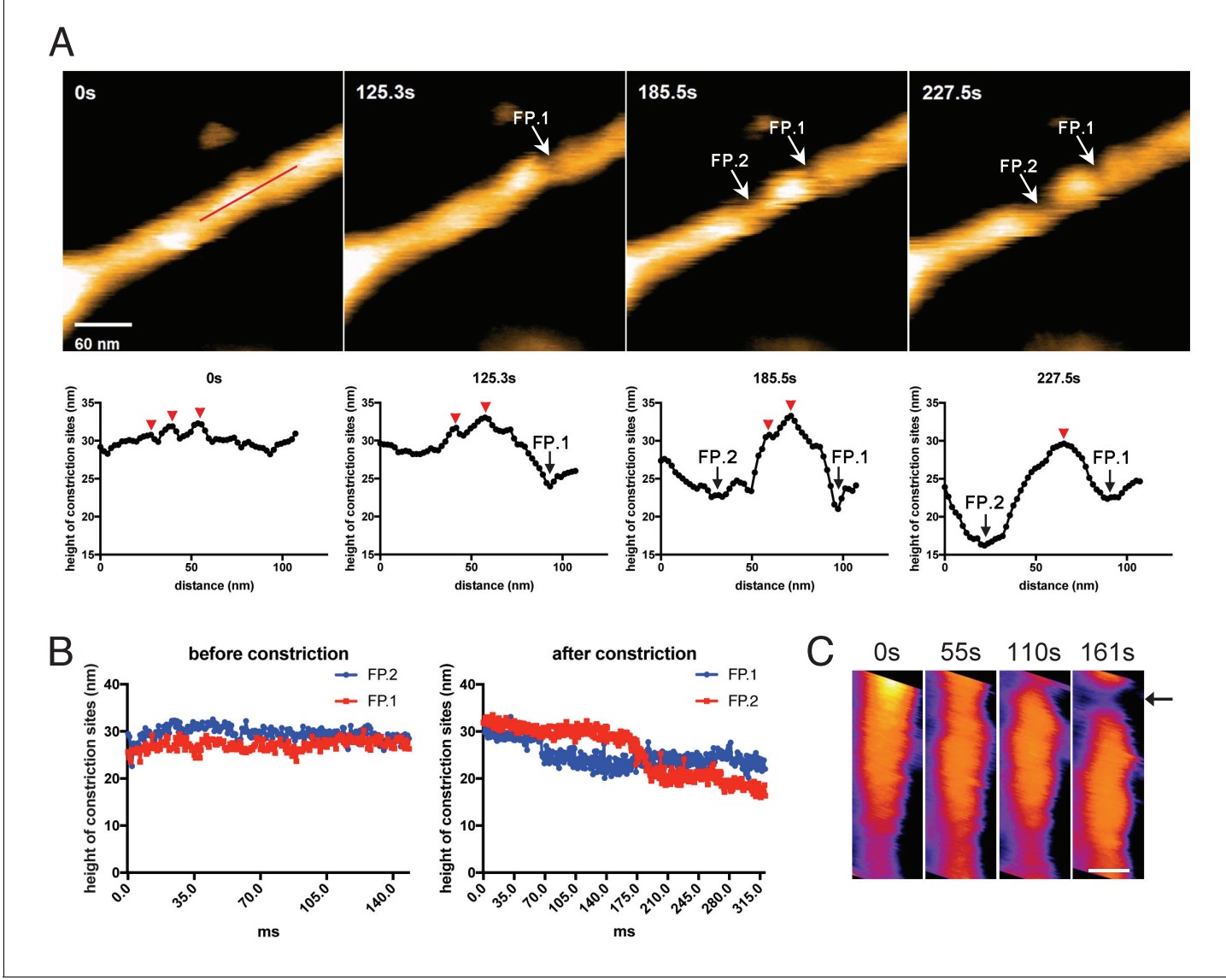

**Figure 4.** Membrane fission occurs at the protein-uncoated regions flanking dynamin-amphiphysin clusters. (**A**) Clips of HS-AFM images captured at 0.42 frames/s showing membrane fission by dynamin-amphiphysin complexes (0 s, 125.3 s, 185.5 s and 227.5 s) in **Video 6**. Membrane fission occurred at flanking regions of a dynamin-amphiphysin cluster. Corresponding height profiles along the red line (shown in the 0 s image) passing through the two fission points (arrows marked with FP.1 and FP.2) are shown below, together with clustered dynamin-amphiphysin helical complexes (red arrowheads). (**B**) Height profiles at fission points (FP.1 and FP.2) over time before (**Video 7**) and after constriction (**Video 6**). The heights of the lipid tubules from the substrate surface were measured at the fission points. (**C**) Clips of HS-AFM images showing clustering dynamin-amphiphysin complexes and membrane constriction at flanking regions of the cluster (arrow). HS-AFM images are shown in pseudo color. Scale bar is 40 nm.
DOI: https://doi.org/10.7554/eLife.30246.020

The following source data is available for figure 4:

**Source data 1.** Measuring height changes of lipid tubules during constriction and fission by dynamin-amphiphysin helices.
DOI: https://doi.org/10.7554/eLife.30246.021

**Source data 2.** Measuring heights of fission points (FP.1 and FP.2) over time before (**Video 7**) and after GTP addition (**Video 6**) for panel B.
DOI: https://doi.org/10.7554/eLife.30246.022

GTPase reaction is responsible for the membrane fission or how many GTPase cycles are required for it.

In this study, we revealed that dynamin-amphiphysin helical complexes are rearranged to form their clusters upon GTP hydrolysis (**Figure 2** and **Figure 3**) and membrane fission occurs at the

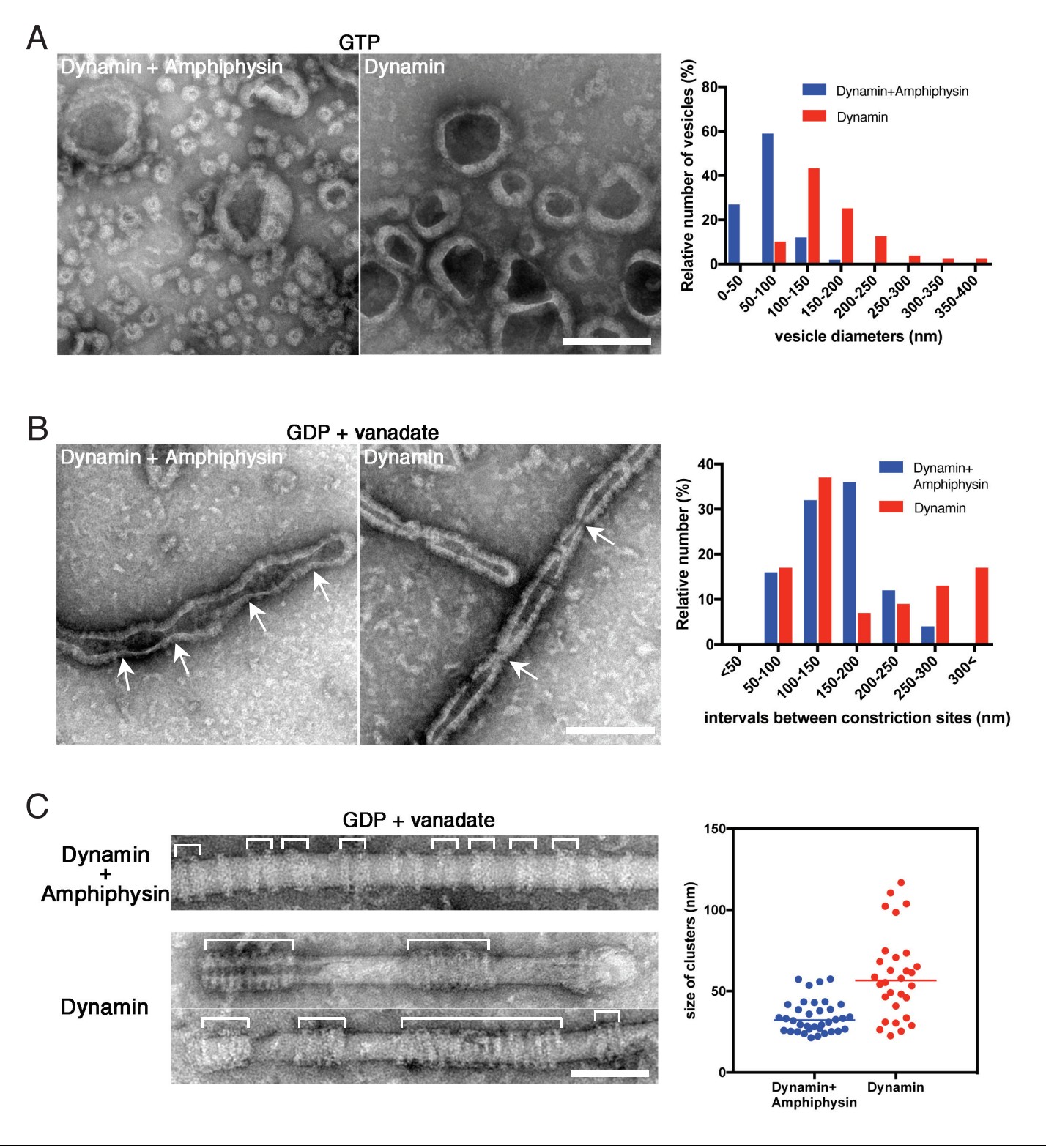

**Figure 5.** Amphiphysin contributes to generation of uniformly-sized vesicles by controlling dynamin-amphiphysin clusters. (**A**) Representative EM images of membrane vesicles generated by dynamin-amphiphysin complexes (Dynamin + Amphiphysin) or dynamin alone (Dynamin) after addition of GTP. Size distribution of generated vesicles are shown in the right panel. The average sizes of vesicles were 70.0 ± 0.6 nm (mean diameter ± s.e.m., n > 30, N = 3) for dynamin-amphiphysin complexes and 204.6 ± 1.1 nm (mean diameter ± s.e.m., n > 45, N = 3) for dynamin alone. Scale bar is 200 nm. (**B**) Representative EM images of membrane constriction induced by dynamin-amphiphysin complexes (Dynamin + Amphiphysin) and dynamin alone
*Figure 5 continued on next page*

*Figure 5 continued*

(Dynamin) in the presence of GDP and vanadate. Distribution of intervals between constriction sites (arrows) are quantified in the right panel. The average intervals of constriction sites induced are 150.3 ± 9.8 nm (mean intervals ± s.e.m., n = 25 from 7 tubes) by dynamin-amphiphysin complexes and 193.5 ± 15.8 nm (mean intervals ± s.e.m., n = 46 from 15 tubes) by dynamin alone. Scale bar is 200 nm. (**C**) Clustering of dynamin-amphiphysin complexes (Dynamin + Amphiphysin) and dynamin alone (Dynamin) on lipid nanotubes in the presence of GDP and vanadate. Clusters of dynamin-amphiphysin helices are indicated (white brackets). Distribution of cluster size were shown as scattered plot in the right panel. Average size of the clusters formed by dynamin-amphiphysin complexes and dynamin alone are 34.2 ± 1.7 nm (mean cluster size ± s.e.m., n = 36 from 7 tubes) and 59.3 ± 4.7 nm (mean cluster size ± s.e.m., n = 30 from 5 tubes) respectively. Scale bar is 100 nm.

DOI: https://doi.org/10.7554/eLife.30246.026

The following source data is available for figure 5:

**Source data 1.** Measuring diameters of vesicles generated by dynamin-amphiphysin complexes or dynamin alone after GTP addition for panel A.

DOI: https://doi.org/10.7554/eLife.30246.027

**Source data 2.** Measuring distances between membrane constriction sites induced by dynamin-amphiphysin complexes and dynamin alone in the presence of GDP and vanadate for panel B.

DOI: https://doi.org/10.7554/eLife.30246.028

**Source data 3.** Measuring size of clusters formed by dynamin-amphiphysin complexes and dynamin alone in the presence of GDP and vanadate for panel C.

DOI: https://doi.org/10.7554/eLife.30246.029

flanking 'protein-uncoated' membrane regions (*Figure 4*). In the 'constrictase' model, dynamin constricts membrane until the membrane neck reaches to the hemi-fission state, which leads to spontaneous membrane fission (*Chen et al., 2004*; *Hinshaw and Schmid, 1995*; *Mears et al., 2007*). However, several lines of evidences are apparently inconsistent with this simple model. For instance, the super-constricted state of dynamin does not constrict the membrane sufficiently enough to reach the hemi-fission state (*Sundborger et al., 2014*) and membrane tension and/or torsion is required to overcome the energy barrier to fission (*Bashkirov et al., 2008*; *Morlot et al., 2012*; *Roux et al., 2006*). In this study, we showed that GTP hydrolysis induces constriction of the dynamin-amphiphysin helices as well as clustering (*Figure 3*). These radial and longitudinal remodeling of the dynamin-amphiphysin helices may give local tension and/or torsion to the membrane tube at the edge of the clusters to drive membrane fission. Alternatively, the dynamin-amphiphysin clusters may serve as a lipid diffusion barrier that causes friction leading to membrane scission (*Simunovic et al., 2017*). Longitudinal rearrangement upon GTP hydrolysis similar to the cluster formation by the dynamin-amphiphysin complexes was also observed in an EM study on the dynamics of dynamin with lipid nanotubes (*Stowell et al., 1999*) and more recently by HS-AFM analyses on dynamics of ΔPRD dynamin (*Colom et al., 2017*), suggesting that the longitudinal rearrangement is an intrinsic property of dynamin during membrane fission.

In our previous studies, we showed that amphiphysin enhances dynamin's GTPase activity in the presence of liposomes (*Takei et al., 1999*; *Yoshida et al., 2004*). In this study, we revealed that amphiphysin may also contributes to effective vesicle biogenesis by controlling the number of constriction sites via cluster formation of dynamin-amphiphysin helices in a long membrane tubule formed in vitro (*Figure 5*). Although precise mechanisms of the cluster size control by amphiphysin remains unclear, amphiphysin could have roles either in determining the number of dynamin-amphiphysin helices comprising the clusters, or in positioning of breakage points in dynamin-amphiphysin helices to induce clustering. Tubular structures have long been known to be present in various synapses, and they are described as 'membrane tubules' (*Heuser and Miledi, 1971*), 'cisternae' (*Heuser and Reese, 1973*), 'synaptic tubules' (*Samorajski et al., 1966*) or 'anastomosing tubules' (*Ekström von Lubitz, 1981*). The tubules are enriched in endocytic proteins including dynamin, synaptojanin, amphiphysin, and endophilin (*Fuchs et al., 2014*; *Takei et al., 1998*), and the presence of the tubules becomes more prominent when synapses are stimulated (*Fuchs et al., 2014*; *Takei et al., 1998*), or when membrane fission is blocked in dynamin 1 K.O. mice (*Ferguson et al., 2007*). These findings strongly suggest that the tubular structures represent endocytic intermediate at which dynamin-amphiphysin-dependent synergic vesicle formation takes place in the synapse. Besides amphiphysin, other BAR domain proteins, endophilin and syndapin, are also implicated in synaptic vesicle recycling (*Dittman and Ryan, 2009*; *Koch et al., 2011*; *Milosevic et al., 2011*). Interestingly, recent study showed that endophilin potently inhibits the dynamin-mediated membrane fission by intercalating dynamin rungs and preventing their trans-interactions

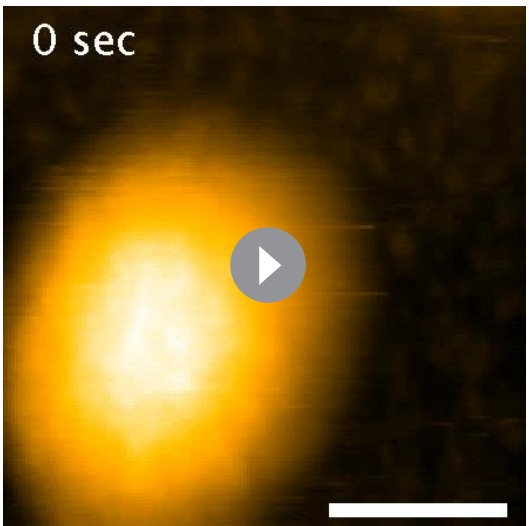

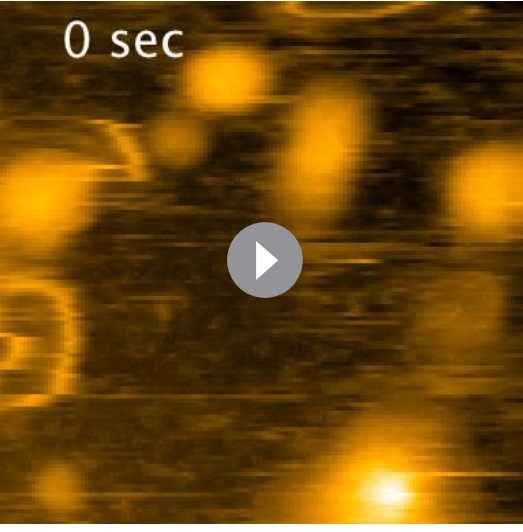

**Video 1.** HS-AFM imaging of a LUV.
DOI: https://doi.org/10.7554/eLife.30246.015

**Video 2.** HS-AFM imaging of a lipid tubules induced from LUVs by dynamin-amphiphysin complexes.
DOI: https://doi.org/10.7554/eLife.30246.016

required for membrane fission (*Hohendahl et al., 2017*). Although the dynamin-mediated membrane fission is also inhibited when an excess of amphiphysin co-assembles with dynamin (*Figure 1—figure supplement 2C*), it is rather stimulatory when the molar ratio of dynamin to amphiphsyin is around 1:1 (*Yoshida et al., 2004*). One of the important future goals of dynamin study would be to clarify regulatory mechanisms by which dynamin alters its interactions with various BAR domain proteins in physiological contexts such as synaptic vesicle recycling.

In conclusion, live imaging analyses using HS-AFM in this study and a study from another group (*Colom et al., 2017*) gave new mechanistic insights into the dynamin-mediated membrane fission. Combinatory approaches using high temporal resolution imaging with HS-AFM and high spatial resolution structural analyses with X-ray crystallography or Cryo-EM will be the most powerful approach in resolving various dynamic membrane remodeling processes in the future.

## Materials and methods

### Purification of dynamin1 and amphiphysin

Human dynamin1 was purified using the method of Warnock et al. with some modification (*Warnock et al., 1996*). Sf9 cells grown in 600 ml of SF-900II SFM (Thermo Fisher Scientific, Waltham, MA) to the cell density of $1 \times 10^6$ cells/ml and the cells were infected with baculoviruses expressing dynamin1. After cultivation of cells at 28°C for 69 hr, the infected Sf9 cells were harvested by centrifugation at $500 \times$ g for 10 min. The cell pellet was resuspended by 1/20 of the culture volume (30 ml) of HCB (Hepes column buffer)100 (20 mM Hepes, 100 mM NaCl, 2 mM EGTA, 1 mM $MgCl_2$, 1 mM DTT, 1 mM PMSF, 1 µg/ml Pepstatin A, 40 µM ALLN, pH 7.2) and cells were sonicated using a sonicator (Advanced-Digital SONIFIER model 250, BRANSON). The cell lysate was mixed with equal volume of HCB0 (20 mM Hepes, 2 mM EGTA, 1 mM $MgCl_2$, 1 mM DTT, 1 mM PMSF, 1 µg/ml Pepstatin A, 40 µM ALLN, pH 7.2) to make HCB50 (20 mM Hepes, 50 mM NaCl, 2 mM EGTA, 1 mM $MgCl_2$, 1 mM DTT, 1 mM PMSF, 1 µg/ml Pepstatin A, 40 µM ALLN, pH7.2) and centrifuged at $210,000 \times$ g for 1 hr at 4°C. Ammonium sulfate was added to the cleared lysate to the 30% saturation and incubated at 4°C for 30 min and centrifuged at $10,000 \times$ g for 10 min to recover the dynamin1 containing fraction in the pellet. The dynamin1 pellet was resuspended with 20 ml of HCB50 and dialyzed against 2L of HCB50 for total 4 hr (2 hr, 2 times) using dialysis membrane (Spectra/Por Dialysis Membrane MWCO: 3500). The dialyzed dynamin1 fraction was applied to Mono Q5/50 GL column (GE healthcare) and bound proteins were eluted stepwise using HCB50, HCB100,

HCB250 and HCB1000 buffers. Purified dynamin1 was recovered in HCB250 fraction and purity was determined by SDS-PAGE (*Figure 1—figure supplement 1A*, Dynamin).

Human amphiphysin was purified following manufacture's instruction (GE Healthcare) with slight modifications. Host bacteria BL21 (DE3) transformed with an expression construct for GST fusions of human amphiphysin (pGEX6P2-HsAMPH) were grown in 1 L of LB medium to the cell density of 0.6– 0.8 (OD 600 nm) at 37°C and then protein expression was induced at 18°C for 12 hr in the presence of 0.1 mM IPTG. The bacterial cells were harvested by centrifugation at 7000 × g for 10 min and cell pellet was resuspended by 1/10 culture volume (100 ml) of Elution/Wash 300 buffer (50 mM Tris-HCl, pH 8.0, 300 mM NaCl). The resuspended cells were sonicated using Advanced-Digital SONI-FIER model 250D (Branson) and centrifuged at 261,000 × g for 30 min at 4°C and cleared lysate was recovered in supernatant. To the cleared lysate, 1/100 culture volume (1 ml in bed volume) of Gluta-thione Sepharose 4B Beads (GE Healthcare) was added and they are mixed using rotating mixer for 1 hr at 4°C. The beads were washed with the Elution/Wash 300 buffer for 5 times in a repeated cycle of centrifugation at 420 × g for 5 min at 4°C followed by mixing with rotator for 5 min at 4°C. The beads with purified GST fusions of amphiphsyin were equilibrated with PreScission Buffer (50 mM Tris-HCl, 150 mM NaCl, 1 mM EDTA, 1 mM DTT, pH 7.0) and GST-tag was removed by PreScission Protease (GE Healthcare) by incubating for 12 hr at 4°C. The purified amphiphysin was recovered by centrifuge (12,000 × g, 5 min at 4°C) using spin column (Ultrafree-Mc, GV 0.22 μm, Millipore) and purity was determined by SDS-PAGE (*Figure 1—figure supplement 1A*, Amphiphysin).

## Preparation of LUVs and lipid nanotubes

Large unilamellar vesicles (LUVs) and lipid nanotubes were prepared as previously described (*Takei et al., 2001*). For LUVs, 70% PS (Cat. No 840032C, Avanti), 10% biotinPE (Avanti) and 20% cholesterol (Avanti) were mixed and, for lipid nanotubes 40% NFA Galactocerebrosides (Sigma C1516), 40% PC (Avanti), 10% PI(4,5)P$_2$ (Calbiochem) and 10% cholesterol (Avanti) were mixed in 250 μl of chloroform in a glass vial (Mighty Vial No.01 4 ml, Maruemu Cat. No 5-115-03). Then chloroform was evaporated using slow-flow nitrogen gas to produce lipid a lipid film on the glass and then completely dried in a vacuum desiccator for 30 min. The dried lipid was rehydrated by water-saturated nitrogen gas followed by addition of 250 μl of filtered 0.3M sucrose for 2 hr at 37°C. The resultant LUVs and lipid nanotubes were passed through 0.4 μm- and 0.2μm- polycarbonate filters respectively 11 times using Avanti Mini

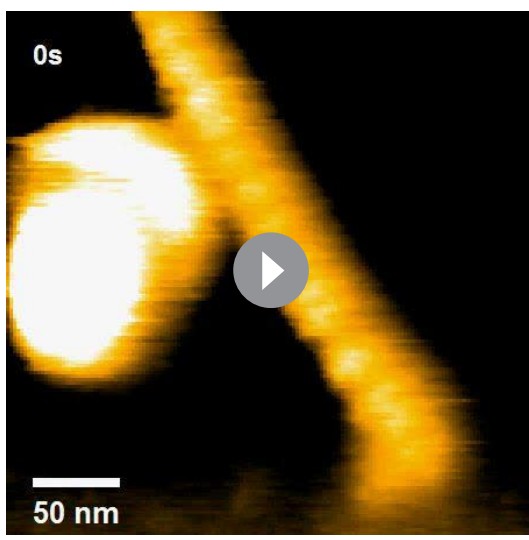

**Video 3.** HS-AFM imaging of cluster formation by dynamin-amphiphysin helical complexes.
DOI: https://doi.org/10.7554/eLife.30246.017

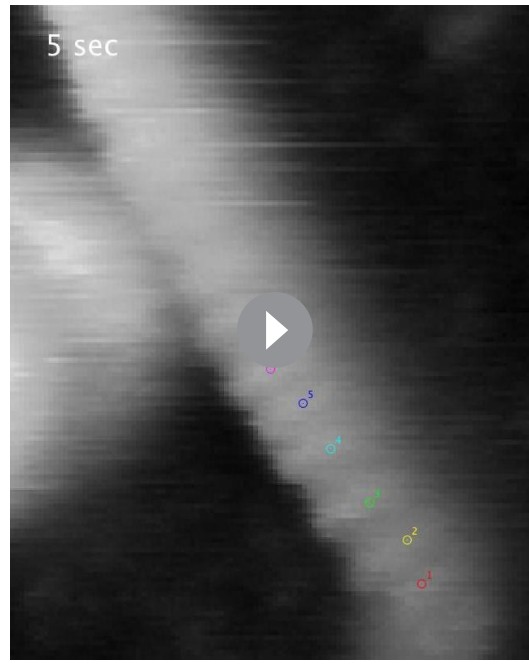

**Video 4.** Particle tracking of dynamin-amphiphysin helical complexes before GTP addition (frames from 5 s to 21 s in *Video 3*).
DOI: https://doi.org/10.7554/eLife.30246.018

extruder. The LUVs and lipid nanotubes (1 mg/ml of final concentration) were stored in dark at 4°C avoiding photooxidation.

## EM imaging of in vitro assay with liposomes and dynamin-amphiphysin complexes

LUVs and lipid nanotubes were diluted to 0.17 mg/ml in cytosolic buffer (25 mM Hepes-KOH, pH 7.2, 25 mM KCl, 2.5 mM Magnesium acetate, 0.1 M K-glutamate, pH 7.4). Dynamin-amphiphysin complexes (1:1 in molar ratio) were diluted to 2.3 µM in the cytosolic buffer. Formvar filmed EM grids were carbon-coated, then glow-discharged. Droplets of the diluted lipids (10 µl each) were prepared on Parafilm and adsorbed on EM grids for 5 min at room temperature. Then the EM grids with lipids were transferred to other droplets of the diluted dynamin-amphiphysin complexes and incubated for 30 min at room temperature in a humid chamber. To see the temporal effect of GTP hydrolysis, the EM grids were transferred to 1 mM of GTP and incubated for various time periods (from 1 s to 10 min). The reaction was terminated by quick removal of the GTP solution by filter paper at room temperature. Alternatively, the EM grids were incubated either GTP, GTPγS, GMP-PNP, GDP plus Vanadate and GDP to analyze GTP hydrolysis transition state structures. The EM grids were negatively stained with filtered 2% uranyl acetate and observed with transmission electron microscope (HITACHI H-7650).

## HS-AFM imaging

All AFM images shown in this article were capture by a laboratory-built HS-AFM in which the amplitude-modulation mode was used. For the HS-AFM imaging, a small cantilever with dimensions of 7 µm long, 2 µm wide, and 90 nm thick was used (Olympus). Its nominal spring constant and resonant frequency were ~0.2 N/m and ~800 kHz in an aqueous solution, respectively. To obtain a sharp tip, an amorphous carbon pillar was grown on the original bird-beak tip of the cantilever by electron beam deposition (EBD) and then sharpened by a plasma etching in an argon environment. The typical radius of the EBD tip was approximately 2 nm after sharpening. For the amplitude-modulation imaging, the cantilever was oscillated with amplitude less than 10 nm under free oscillation condition and the set-point was set at ~90% of the free oscillation amplitude. For HS-AFM imaging of liposomes and dynamin-amphiphysin complexes with lipid tubules or nanotubes, we used mica covered with carbon film. After coating a freshly

**Video 5.** Particle tracking of dynamin-amphiphysin helical complexes after GTP addition (frames from 38 s to 124 s in *Video 3*).
DOI: https://doi.org/10.7554/eLife.30246.019

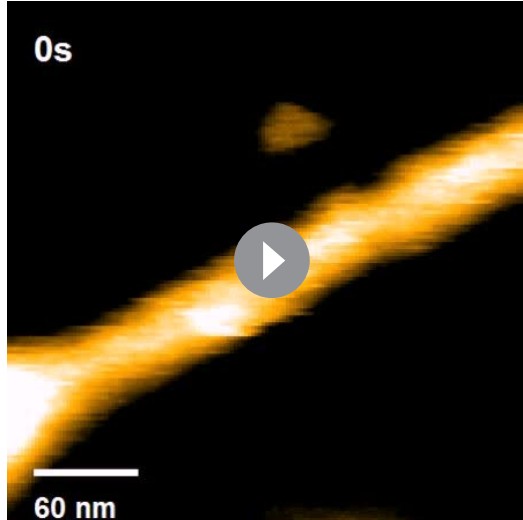

**Video 6.** HS-AFM imaging of constriction and fission of lipid tubules by dynamin-amphiphysin complexes.
DOI: https://doi.org/10.7554/eLife.30246.023

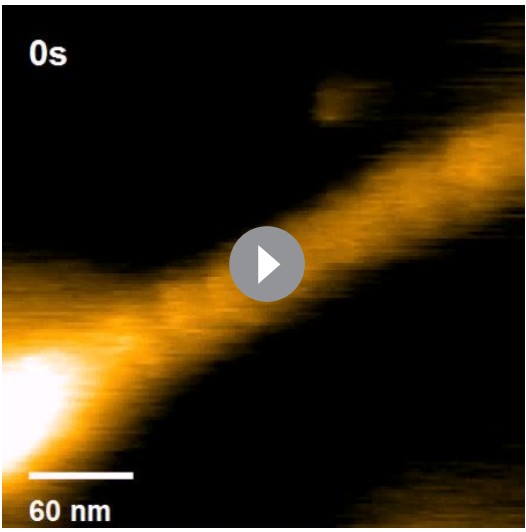

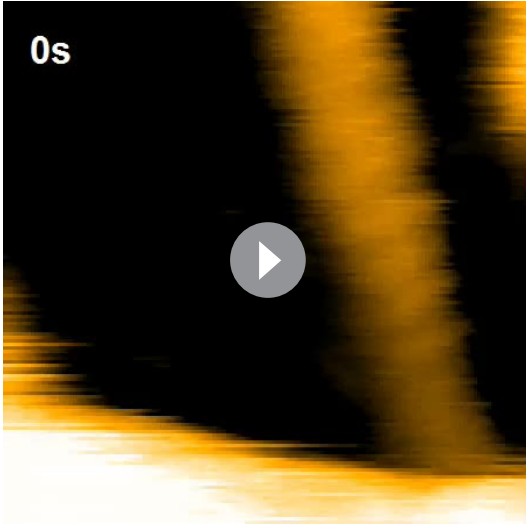

**Video 7.** HS-AFM imaging of lipid tubules before constriction and fission by dynamin-amphiphysin complexes.
DOI: https://doi.org/10.7554/eLife.30246.024

**Video 8.** HS-AFM imaging of constriction and fission of lipid tubules by dynamin-amphiphysin complexes.
DOI: https://doi.org/10.7554/eLife.30246.025

cleaved mica surface with carbon film, hydrophilic treatment was carried out by a grow discharge. The liposomes (0.17 mg/ml) were deposited on the hydrophilic mica surface and incubated for 5 min at room temperature followed by deposition of proteins (0.6 µM of dynamin1 and amphiphysin) for 30 min at room temperature. After the incubation, the sample was thoroughly washed by cytosolic buffer to remove excess liposomes and proteins. After the washing, the cantilever tip was approached and the imaging was performed under the buffer.

## Quantitative data analyses of EM and HS-AFM images

The EM and HS-AFM images were randomly captured to avoid data manipulation and representative images were shown in all the figures. The average pitch between the dynamin-amphiphysin helices in EM images (*Figure 2*) and HS-AFM images (*Figure 3*), diameter of vesicles, intervals between constriction sites and size of clusters generated by either dynamin-amphiphysin complex or dynamin (*Figure 5*), were all measured by FIJI (*Schindelin et al., 2012*). Experimental data were statistically analyzed using Excel (Microsoft) or Prism 7 (GraphPad software).

## Acknowledgements

The authors thank Dr. Harvey McMahon (MRC-LMB) for sharing valuable reagent (NFA Galactocerebrosides). The authors also thank Dr. Aurélien Roux and Dr. Adai Colom (University of Geneva), Dr. Lorena Redondo-Morata (INSERM U1006/Aix-Marseille Université) and Dr. Masatoshi Ichikawa (Kyoto University) for their critical reading of the manuscript. This work was supported by JST/CREST program (#JPMJCR13M1) (to TA and KT). This work was also supported in part by JSPS KAKENHI Grant Numbers JP15H03540 (to TU), MEXT KAKENHI Grant Numbers JP16H00830 and JP16H00758 (to TU).

## Additional information

### Funding

| Funder | Grant reference number | Author |
| --- | --- | --- |
| Japan Science and Technology Agency | JPMJCR13M1 | Toshio Ando Kohji Takei |

| Japan Society for the Promotion of Science | JP15H03540 | Takayuki Uchihashi |
|---|---|---|
| Ministry of Education, Culture, Sports, Science, and Technology | JP16H00830 | Takayuki Uchihashi |
| Ministry of Education, Culture, Sports, Science, and Technology | JP16H00758 | Takayuki Uchihashi |

The funders had no role in study design, data collection and interpretation, or the decision to submit the work for publication.

### Author contributions

Tetsuya Takeda, Conceptualization, Resources, Data curation, Formal analysis, Supervision, Validation, Investigation, Visualization, Methodology, Writing—original draft, Project administration, Writing—review and editing; Toshiya Kozai, Huiran Yang, Daiki Ishikuro, Kaho Seyama, Yusuke Kumagai, Data curation, Formal analysis, Investigation, Methodology; Tadashi Abe, Resources, Data curation, Formal analysis, Investigation, Methodology, Writing—review and editing; Hiroshi Yamada, Conceptualization, Writing—review and editing; Takayuki Uchihashi, Conceptualization, Data curation, Software, Formal analysis, Supervision, Funding acquisition, Validation, Investigation, Methodology, Project administration, Writing—review and editing; Toshio Ando, Kohji Takei, Conceptualization, Supervision, Funding acquisition, Methodology, Project administration, Writing—review and editing

### Author ORCIDs

Tetsuya Takeda (iD) http://orcid.org/0000-0002-3183-6551

### Decision letter and Author response

Decision letter https://doi.org/10.7554/eLife.30246.032
Author response https://doi.org/10.7554/eLife.30246.033

## Additional files

### Supplementary files

• Transparent reporting form
DOI: https://doi.org/10.7554/eLife.30246.030

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
