## [Decision Letter]

Thank you for submitting your article "Dynamic clustering of dynamin-amphiphysin rings regulates membrane constriction and fission coupled with GTP hydrolysis" for consideration by *eLife*. Your article has been favorably evaluated by Vivek Malhotra (Senior Editor), a Reviewing Editor, and three reviewers. The following individuals involved in review of your submission have agreed to reveal their identity: Aurélien Roux (Reviewer #1) and Simon Scheuring (Reviewer #3).

The reviewers have discussed the reviews with one another and the Reviewing Editor has drafted this decision to help you prepare a revised submission.

Overall, your findings are of interest, but the reviewers found some aspects of the mechanisms of amphiphysin/dynamin clustering confusing and the fission model also raised contradictory arguments.

We invite you to submit a revised manuscript provided you can address the following major points satisfactorily.

Major points:

1) Explain in better terms how amphiphysin participates in the clustering of dynamin rings. In particular, the reviewers were concerned by the distinction between helices and rings: do dynamin and amphiphysin polymerize as ring, that are then clustered, or is the helical pitch of dynamin helix changed by amphiphysin? Another possibility suggested by the reviewers is that the dynamin helix could be "broken" into rings by amphiphysin, which are then further clustered.

2) Regarding your model of fission: It is unclear how your model differs from previously proposed models. The reviewers agree that both the figure and the text should be changed to explain better the role of amphiphysin in clustering of rings/helices of dynamin.

3) Some of the authors have shown previously that the stimulatory action of amphiphysin on dynamin is curvature-dependent, and most probably stoichiometry dependent. The reviewers thus ask to clarify what is the stoichiometry of amphiphysin to dynamin (concentration ratios), and to perform similar EM experiment as shown in Figure 1 (No AFM experiment are requested) with 2-3 different stoichiometries to test how this clustering behavior as well as the fission efficiency is affected by amphiphysin/dynamin ratio.

*Reviewer #1:*

The manuscript by Takeda et al. focuses on the dynamics of dynamin/amphiphysin copolymer upon GTP hydrolysis. The authors use a combination of beautiful electron microscopy techniques and high-speed AFM to show that upon GTP hydrolysis, adjacent rings of dynamin cluster by pairs or more. They also show at the nanometric scale that fission occurs in between those pairs rings. The AFM images show that the pairing is due to dynamic clustering of rings rather than depolymerization/polymerization of rings in another state. Overall, the paper is very convincing and balanced in its arguments. Moreover, it shows how a large structure such as the dynamin helix can dynamically change conformation under the influence of partners and nucleotides. It is also consistent with a recent publication on dynamin, also using high-speed AFM (Colom et al. PNAS 2017) and showing similar dynamics of the dynamin helix, but with less precision. I thus strongly recommend publication.

I however have one experiment to suggest. I guess the authors have used a stoichiometry of 1:1 to generate the dynamin:amphiphysin complex (I could not find the information at second read, but remembered it was 1:1). I was wondering how the results would evolved as a function of the stoichiometry: does increasing amphiphysin's ratio increases clustering dynamics? I think EM data on this would suffice, and this experiment is not essential.

*Reviewer #2:*

The authors have used a combination of high-resolution microscopy, negative staining electron microscopy and high speed AFM to study the organization of amphiphysin and dynamin on membranes. It was previously shown that amphiphysin amplified the enzymatic activity of dynamin and that dynamin-induced scission during endocytosis was perturbed when amphiphysin was deleted. But, the mechanism behind this "boosting" activity is still unclear. Thus, the question is still open. The most interesting result of the paper is the observation of the non-regular helix formed by amphiphysin + dynamin when GTP is present. However, I globally find that this paper is not rigorous enough, with no clear contribution to the understanding of the boosting effect and unjustified conclusions. I thus do not recommend publication of the paper in *eLife*.

1) In their paper, the authors generally describe the arrangement of the proteins on soft or rigid tubes as rings rather than as helices. However, although these proteins form rings in solution (Figure 1—figure supplement 1 on this point is not different from those from Takei et al. (NCB, 1999)…), it is well accepted they form helices on tubules, either independently or together. This point is particularly important for the discussion of the effect of GTP hydrolysis. Indeed, rings refer to independent structures, which might interact laterally, while helices are continuous structures and their change of organization would be completely different.

2) The authors observe the reorganization of the helices on rigid lipid nanotubes, as well as on soft tubules, in the presence of GTP or GDP + Vanadate. They describe their observation as a "clustering" of the rings, which is quite misleading. Indeed, the regularity of the protein organization is perturbed, with a non-even distribution of the helices; however clustering again would be more adapted to rings than to helices. Moreover, with dynamin alone, a change of the helix pitch is also observed (Figure 5) and in this case, larger "clusters" than in the presence of amphiphysin are obtained. Unfortunately, the authors do not discuss the structural origin of this "clustering" effect and of the effect of amphiphysin on the organization of the dynamin helices during GTP hydrolysis.

3) From HS-AFM, the authors observe the dynamics of tubule constriction. The authors did not discuss the very long time required for this constriction, a few minutes after GTP addition. Moreover, they do not observe fission, which they claim is due to the strong attachment of the tubules to the substrate. They conclude that constriction occurs on the bare membrane near the edge of the dynamin/amphiphysin "cluster", and from that extrapolate that scission would occur at the same place, although not observed here. Moreover, some constriction seems also to occur under the proteins since the height decreases (Figure 4, 227s), but this point is not discussed. The paper from Colom et al. (PNAS, 2017) that uses HS-AFM on dynamin alone has shown that constriction occurs under the dynamin helix; it would be insightful to discuss the results in comparison with this paper. Eventually, the statistics is rather poor (1 or 2 examples) and the quality of the HS-AFM data is not very good.

4) The authors show that the size of the vesicles generated by fission of the tubules is smaller when amphiphysin is present as compared with dynamin alone. They suggest that the vesicle size is related to the distance between "ring clusters" on tubules. However, in vivo amphiphysin is involved with dynamin in clathrin-mediated endocytosis, at the neck of spherical buds, and not on tubules. It is thus exaggerating to extrapolate from the observation on the size of vesicles generated in vitro from the scission of tubules, conclusions on the control of vesicle size during clathrin-mediated endocytosis.

5) Overall, it is not clear for me what insight is brought by the "clusterase model" to the fission mechanism. In fact, no real mechanism is proposed that explains how amphiphysin facilitates fission, rather a very vague list of possible effects (Discussion) without solid physical basis.

*Reviewer #3:*

The paper "Dynamic clustering of dynamin-amphiphysin rings regulates membrane constriction and fission coupled with GTP hydrolysis" presents EM and HS-AFM data of dynamin-amphiphysin complexes on lipid tubules and nanorods providing insights into the constriction/fission mechanism of the proteins.

The topic is timely, the data is of highest quality, and the insights of large interest in fields ranging from biophysics to cell biology.

There are essentially 4 major findings:

1) Dynamin-amphiphysin rings cluster upon GTP hydrolysis.

2) Membrane constriction occurs at protein uncoated regions (between clusters).

3) GTP hydrolysis is required and sufficient for clustering.

4) Amphiphysin controls the cluster size.

Altogether, the authors propose a "clusterase" model as the new functional mechanism of action. The authors put the new finding into cellular context (Discussion, third paragraph) which is very nice.

Evidence is presented for all four findings. Some aspects should be strengthened:

Clustering: what happens between the clusters? Either the dynamin helix would have a huge pitch (that always occurs on the invisible backside of tubule – very unlikely) or the helix is broken when clusters form. In this respect "clustering" would be "breakage". I am puzzled why the authors do not comment about this. It might be a most important aspect of their findings that amphiphysin is a dynamin helix breaker.

Clustering regulation by amphiphysin: The authors compare +amphiphysin with pure dynamin tubules. Would it be possible to test the hypothesis by addition of low (different) amounts of amphiphysin and analyze by EM cluster size?

Figure 6 is missing? As a consequence I am not understanding the proposed fission process.

Overstatement:

Abstract and Introduction, last paragraph: The use of EM and AFM is in no way "our new approaches" or "a novel approach combining". There are papers since the 90s that combine EM and AFM, and also papers combining EM and HS-AFM (e.g., Colom et al. 2017).

Statistics:

The authors give numbers to the 1/10 of the Å (e.g. helical pitch), which does not make sense.

[Editors' note: further revisions were requested prior to acceptance, as described below.]

Thank you for resubmitting your work entitled "Dynamic clustering of dynamin-amphiphysin helices regulates membrane constriction and fission coupled with GTP hydrolysis" for further consideration at *eLife*. Your revised article has been favorably evaluated by Vivek Malhotra (Senior Editor), a Reviewing Editor, and three reviewers.

The manuscript has been improved but there are some remaining issues that need to be addressed before acceptance, as outlined below:

1) The model does not make much sense in physical terms, and has raised questions from the reviewers (see below). We thus request you to remove the model figure and the text describing the model, as it does not preclude the essential findings of the authors.

2) Since a recent *eLife* paper has been published on dynamin-endophilin structure and inhibition (Hohendahl et al. Life 2017), the results of which are consistent with the stoichiometric experiments added by the authors of this manuscript, the reviewers request that you cite this paper and comment it in the discussion part.

*Reviewer #1:*

The authors have fully answered my questions, and have improved the manuscript by providing data with stoichiometric experiments with amphiphysin and dynamin. They clarified the points asked by other reviewers. I thus recommend this paper for publication.

*Reviewer #2:*

I am satisfied with the answers of the authors related to helix versus ring and the new experiments showing that when the relative amphiphysin:dynamin ratio increases, scission is less efficient. This is online with the paper from Hohendahl A, et al. (2017) in *eLife* for endophilin. I think this paper should be cited.

Nevertheless, I still have a serious problem with the last part of the Discussion and the model, which is totally handwaving with no physical consistency. This is only a cartoon that takes up the images of the paper, but without solid mechanism. For instance, on Figure 6C, how is it possible to create torsion in a fluid membrane and why would tension increase? If constriction occurs under the amphiphysin-dynamin ring, why would the tubule radius decrease next to the coat, and not increase, in particular if there is some friction under the coat that tend to perturb lipid diffusion? I don't understand how the friction model from Simunivic et al. (2017) could work here. In the absence of clear physical description that justifies these possible mechanisms, I wonder what is so new in this manuscript and groundbreaking compared to the PNAS paper from Colom et al. that would justify publication in *eLife*.

*Reviewer #3:*

The author's responses to the first round of reviewers' comments are careful and additional experiments have been performed. Notably, the experiments with varying dynamin:amphiphysin ratios clarify several questions.

The manuscript reads well, and overstatements and vague statements have been removed. In my view the paper provides valuable and novel data showing "Dynamic clustering of dynamin-amphiphysin helices", a timely and important topic, and as such deserves publication in *eLife* in this revised and improved form.

---

## [Author Response]

Major points:1) Explain in better terms how amphiphysin participates in the clustering of dynamin rings. In particular, the reviewers were concerned by the distinction between helices and rings.

We thank reviewers for raising this issue and we apologize for misleading terminology. We also think that dynamin or dynamin-amphiphysin complexes form continuous “helices” but not stacks of “rings” on lipid templates. Indeed, we meant with the terms “washer ring” or “ring” for the split washer-like structure that corresponds to one helical turn. To avoid any confusion, we integrated those terms into “helix (helices)” or “helical” for both dynamin- and dynamin/amphiphysin-complexes on lipid templates in the revised manuscript. However, we decided to keep the term “ring” to describe the lipid-free dynamin-amphiphysin complexes in solution, since it sounds more appropriate.

Do dynamin and amphiphysin polymerize as ring, that are then clustered, or is the helical pitch of dynamin helix changed by amphiphysin? Another possibility suggested by the reviewers is that the dynamin helix could be "broken" into rings by amphiphysin, which are then further clustered.

We think that amphiphysin and dynamin copolymerize to form helices (not rings) in the absence or presence of GTP. Upon GTP hydrolysis, the dynamin-amphiphysin helices form clusters consisting of a few helical turns, and generate protein-uncoated “gap” regions between the clusters. As the reviewer #3 suggested, the helices may be broken at the gap regions, because the clusters are often formed so far apart that the helices are unlikely to stay connected. We added a sentence about the possible helix break in the Discussion (Discussion, third paragraph) and modified the model (Figure 6) in the revised manuscript.

2) Regarding your model of fission: It is unclear how your model differs from previously proposed models. The reviewers agree that both the figure and the text should be changed to explain better the role of amphiphysin in clustering of rings/helices of dynamin.

Since dynamin-amphiphysin complexes form smaller clusters compared to those formed by dynamin alone (Figure 5), amphiphysin may have roles either in determining the cluster size (i.e. number of helices comprising a cluster), or in positioning of break points in the helices. Since we could not discriminate these two possibilities with the current dataset, we discussed these possible roles of amphiphysin in the Discussion section of the revised manuscript.

3) Some of the authors have shown previously that the stimulatory action of amphiphysin on dynamin is curvature-dependent, and most probably stoichiometry dependent. The reviewers thus ask to clarify what is the stoichiometry of amphiphysin to dynamin (concentration ratios), and to perform similar EM experiment as shown in Figure 1 (No AFM experiment are requested) with 2-3 different stoichiometries to test how this clustering behavior as well as the fission efficiency is affected by amphiphysin/dynamin ratio.

We thank Aurélien Roux (Reviewer #1) and Simon Schering (Reviewer #3) for suggesting this experiment. In this study, 1:1 ratio of dynamin and amphiphysin (2.3 𝜇M each in final concentration) was mixed for all the experiments using the dynamin and amphiphysin complexes. Following reviewers’ suggestions, we examined stoichiometry dependency of dynamin and amphiphysin in the ring complex formation (Figure 1—figure supplement 2, A), liposome tubulation (Figure 1—figure supplement 2, B), fission efficiency (Figure 1—figure supplement 2, C) and the clustering behavior on lipid nanotubes (Figure 2—figure supplement 1).

The ring complex formation and liposome tubulation by dynamin and amphiphysin were not stoichiometry dependent (Figure 1—figure supplement 2, 1:0.5, 1:1 and 1:2). In contrast, fission activity by the dynamin-amphiphysin complexes was stoichiometry dependent: it was most efficient at 1:0.5 and 1:1 ratios and less efficient at 1:2. The reduced membrane fission at the 1:2 ratio was probably caused by the less stimulatory activity of amphiphysin on dynamin’s GTPase activity at this ratio (Yoshida et al., EMBO J 2004). Interestingly, clustering behavior of the dynamin-amphiphysin complexes on the lipid nanotubes were not stoichiometry dependent and clusters with a few helical turns were formed in the presence of GDP and vanadate mimicking GDP- Pi state (Figure 2—figure supplement 1, 1:0.5, 1:1 and 1:2). Taken all these results together, we conclude that amphiphysin contributes to both clustering behavior and GTPase stimulation of the dynamin-amphiphysn helical complexes, but the GTPase stimulation is more stoichiometry sensitive. We have added sentences in the main text (subsection “GTP hydrolysis is required and sufficient for membrane constriction by dynamin-amphiphysin complexes”, first paragraph; subsection “GTP hydrolysis induces clustering of dynamin-amphiphysin helices”, last paragraph) and figure legends for Figure 1—figure supplement 2 and Figure 2—figure supplement 1.

[Editors' note: further revisions were requested prior to acceptance, as described below.]

1) The model does not make much sense in physical terms, and has raised questions from the reviewers. We thus request you to remove the model figure and the text describing the model, as it does not preclude the essential findings of the authors.

The model figure (Figure 6) and its legend has been removed. The text describing the model were removed and/or modified (Abstract; Discussion, third paragraph).

2) Since a recent eLife paper has been published on dynamin-endophilin structure and inhibition (Hohendahl et al. Life 2017), the results of which are consistent with the stoichiometric experiments added by the authors of this manuscript, the reviewers request that you cite this paper and comment it in the discussion part.

The recent *eLife* paper (Hohendahl et al. *eLife* 2017) has been cited and commented in the Discussion part (fourth paragraph).